# Rocket Lab Mission to Venus

Richard French [1,*], Christophe Mandy [1], Richard Hunter [1], Ehson Mosleh [1], Doug Sinclair [1], Peter Beck [1], Sara Seager [2,3,4], Janusz J. Petkowski [2], Christopher E. Carr [5], David H. Grinspoon [6], Darrel Baumgardner [7,8] and on behalf of the Rocket Lab Venus Team [†]

1. Rocket Lab, 3881 McGowen Street, Long Beach, CA 90808, USA
2. Department of Earth, Atmospheric and Planetary Sciences, Massachusetts Institute of Technology, 77 Massachusetts Avenue, Cambridge, MA 02139, USA
3. Department of Physics, Massachusetts Institute of Technology, 77 Massachusetts Avenue, Cambridge, MA 02139, USA
4. Department of Aeronautics and Astronautics, Massachusetts Institute of Technology, 77 Massachusetts Avenue, Cambridge, MA 02139, USA
5. School of Aerospace Engineering and School of Earth and Atmospheric Sciences, Georgia Institute of Technology, Atlanta, GA 30332, USA
6. Planetary Science Institute, 1700 East Fort Lowell, Suite 106, Tucson, AZ 85719, USA
7. Droplet Measurement Technologies, LLC, 2400 Trade Centre Ave, Longmont, CO 80503, USA
8. Cloud Measurement Solutions, LLC, 415 Kit Carson Rd., Unit 7, Taos, NM 87571, USA
* Correspondence: r.french@rocketlabusa.com
† Collaborators/Membership of the Group/Team Name is provided in the Acknowledgments.

**Abstract:** Regular, low-cost Decadal-class science missions to planetary destinations will be enabled by high-ΔV small spacecraft, such as the high-energy Photon, and small launch vehicles, such as Electron, to support expanding opportunities for scientists and to increase the rate of science return. The Rocket Lab mission to Venus is a small direct entry probe planned for baseline launch in May 2023 with accommodation for a single ~1 kg instrument. A backup launch window is available in January 2025. The probe mission will spend about 5 min in the Venus cloud layers at 48–60 km altitude above the surface and collect in situ measurements. We have chosen a low-mass, low-cost autofluorescing nephelometer to search for organic molecules in the cloud particles and constrain the particle composition.

**Keywords:** Venus; Rocket Lab; autofluorescing nephelometer; small spacecraft; small launch vehicle

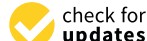



## 1. Introduction

Rocket Lab has made the engineering and financial commitment to fly a private mission to Venus, with a goal of launching in 2023, to help answer the question "Are we alone in the universe?". The specific goals of Rocket Lab's mission are to:

1. Search for habitable conditions and signs of life in Venus's cloud layer;
2. Mature the interplanetary Photon spacecraft;
3. Demonstrate high-performance, low-cost, fast-turnaround deep space entry mission delivering Decadal-class science with small spacecraft and small launch vehicles;
4. Take the first step in a campaign of small missions to better understand Venus.

The baseline mission is planned for launch in May 2023 on Electron from Rocket Lab's Launch Complex 1 (LC-1) with a backup launch opportunity in January 2025. The launch opportunity will be selected to allow for a Trans-Venus Injection (TVI) on 24 May 2023, after sequential phasing orbits around earth and a lunar gravity assist, as was demonstrated on Rocket Lab's successful Cislunar Autonomous Positioning System Technology Operations and Navigation Experiment (CAPSTONE) mission for NASA [1]. The mission will follow a hyperbolic trajectory with the high-energy Photon performing as the cruise stage and then deploying a small probe into the Venus atmosphere for the science phase of the mission. In

this paper, we describe the Photon spacecraft designed for launch on the Electron small launch vehicle (Section 2) followed by the discussion of the spacecraft trajectory (Section 3) and the atmospheric probe itself (Section 4). Section 5 summarizes the probe's concept of operations and the science phase sequence of events. In Section 6, we briefly summarize the science objectives and science instrumentation of the 2023 Rocket Lab mission.

## 2. The Photon Spacecraft

The high-energy Photon (Figure 1), developed by Rocket Lab for the NASA CAP-STONE mission that successfully launched to the moon in June 2022 and also being matured for the NASA Escape and Plasma Acceleration and Dynamics Explorers (ESCAPADE) mission launching to Mars in 2024, is a self-sufficient small spacecraft capable of long-duration interplanetary cruise [2].

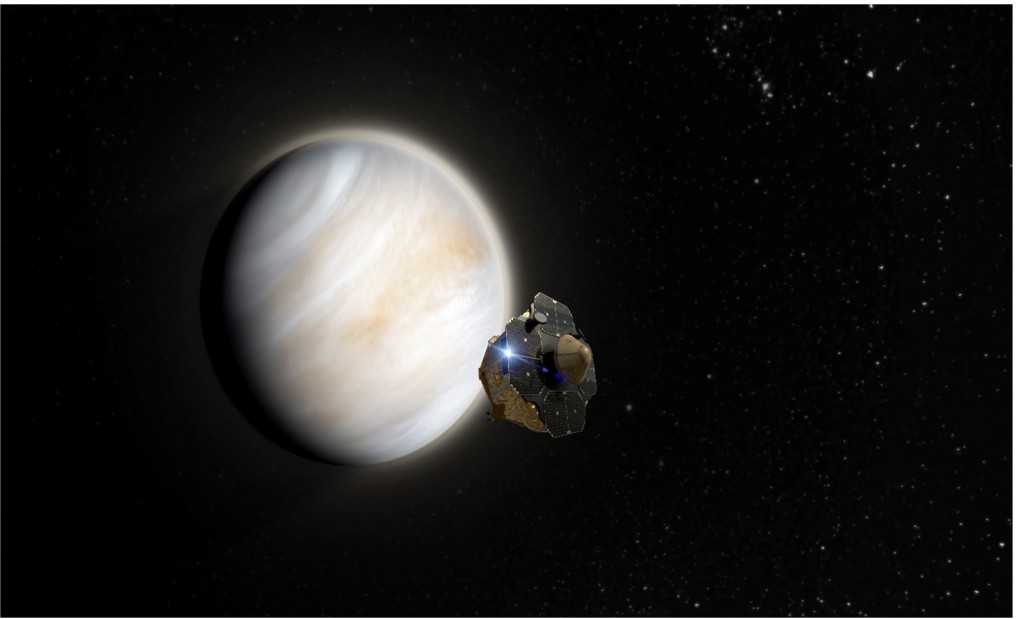

**Figure 1.** Rocket Lab's Electron-launched private mission to Venus will deploy a small probe from a high-energy Photon.

The high-energy Photon's power system is conventional, using photovoltaic solar arrays and lithium-polymer secondary batteries. The attitude control system includes star trackers, sun sensors, an inertial measurement unit, reaction wheels, and a cold-gas reaction control system (RCS). S-band or X-band RF ranging transponders support communications with the Deep Space Network (DSN) or with commercial networks and enable traditional deep space radiometric navigation methods. A Global Position System (GPS) receiver is used for navigation near Earth. $\Delta V$ greater than 3 km/s is provided by a storable, re-startable bi-propellant propulsion system called Hyper Curie using electric pumps to supply pressurized propellant to a thrust vector-controlled engine. The propellant tanks achieve high propellant mass fraction and can be scaled to meet mission-specific needs.

The high-energy Photon (Figure 2) is designed for launch on Electron (Figure 3), Rocket Lab's dedicated small launch vehicle. Electron can lift up to 300 kg to a 500 km orbit from either of two active, state-of-the-art launch sites: LC-1 on the Mahia Peninsula in New Zealand and Launch Complex 2 on Wallops Island, Virginia. Electron is a two-stage launch vehicle with a Kick Stage, standing at 18 m tall with a diameter of 1.2 m and a lift-off mass of ~13,000 kg. Electron's engine, the 25 kN Rutherford, is fueled by liquid oxygen and kerosene fed by electric pumps. Rutherford is based on an entirely new propulsion cycle that makes use of brushless direct current electric motors and high-performance lithium-polymer batteries to drive impeller pumps. Electron's Stage 1 uses nine Rutherford engines while Stage 2 requires just a single Rutherford vacuum engine. Rutherford is the first

oxygen/hydrocarbon engine to use additive manufacturing for all primary components, including the regeneratively cooled thrust chamber, injector pumps, and main propellant valves. All Rutherford engines on Electron are identical, except for a larger expansion ratio nozzle on Stage 2 optimized for performance in near-vacuum conditions. The high-energy Photon replaces the Kick Stage for Electron missions beyond low Earth orbit (LEO).

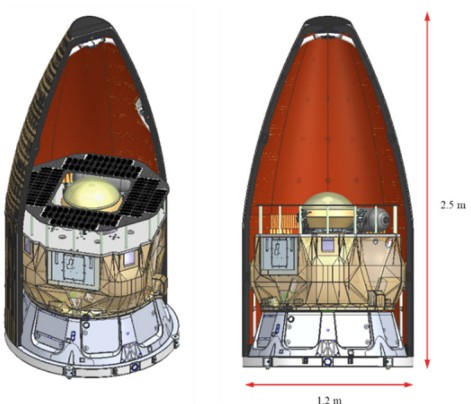

**Figure 2.** High-energy Photon and small Venus entry probe inside Electron's fairing.

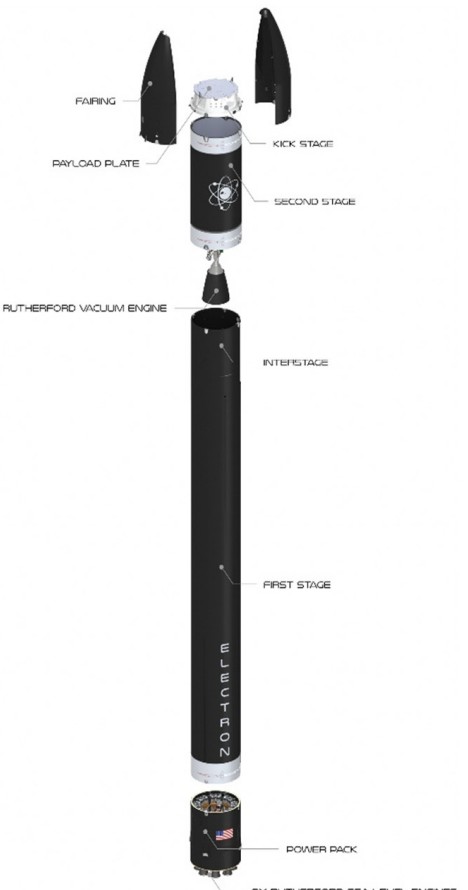

**Figure 3.** Electron small launch vehicle.

### 3. Trajectory

Electron first delivers high-energy Photon to a circular parking orbit (Figure 4) around Earth at roughly 165 km. After separating from Electron's Stage 2, high-energy Photon performs preprogrammed burns to establish a preliminary elliptical orbit of 250 km by ~1200 km. High-energy Photon then performs a series of burns through increasingly

elliptical orbits, each time raising the apogee altitude while maintaining a nearly constant perigee, reaching a maximum apogee of roughly 70,000 km. Breaking the departure across multiple maneuvers is an efficient approach to Earth escape. By holding burns close to perigee and limiting their duration, propulsive energy is efficiently spent raising apogee while avoiding the burn losses associated with long duration maneuvers. Each phasing maneuver is followed by a planned number of phasing orbits at the new apogee altitude. Phasing orbits provide time for on-orbit navigation, maneuver reconstruction and planning, propulsion system calibration, and conjunction screening. Each planned maneuver includes contingency options to mitigate conjunction events or missed maneuvers. After the nominal apogee raising maneuvers are performed, a final injection burn is executed to place high-energy Photon on the escape trajectory. Trajectory correction maneuvers (TCMs) using the Hyper Curie engine or integrated RCS are used to make fine adjustments to the trajectory and target the appropriate entry interface.

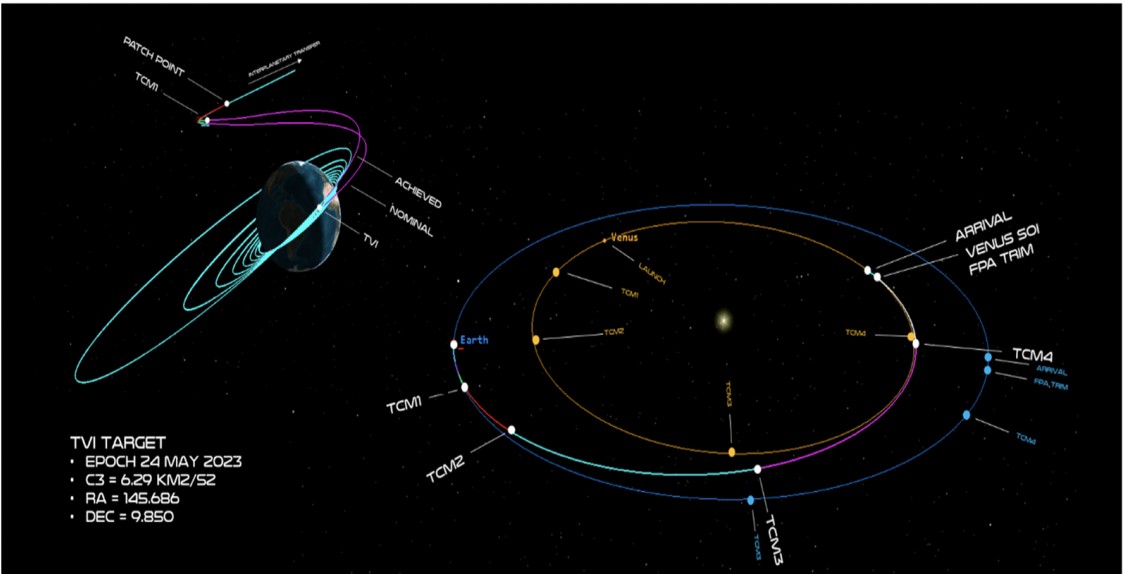

**Figure 4.** Phasing orbits approach to escape trajectory and typical trajectory correction maneuvers are used to target entry interface at Venus.

In October 2023, after the cruise phase (Figure 5), high energy Photon will target an entry interface to deploy a small (~20 kg) probe directly into the atmosphere with an entry flight path angle (EFPA) between −10 and −30 degrees, with a baseline of −10 degrees. The probe communicates direct-to-Earth through an S-band communications link with a hemispherical antenna returning science data captured during the descent and stored on board. The entry interface will be selected to satisfy science objectives (night-side entry and latitude targeting), Earth communication geometry, and other factors. The EFPA will be selected based on an analysis of the entry and descent timeline, the integrated heat load and required thermal protection system (TPS) thickness, probe acceleration (g-loading) limits, navigation precision, and other factors.

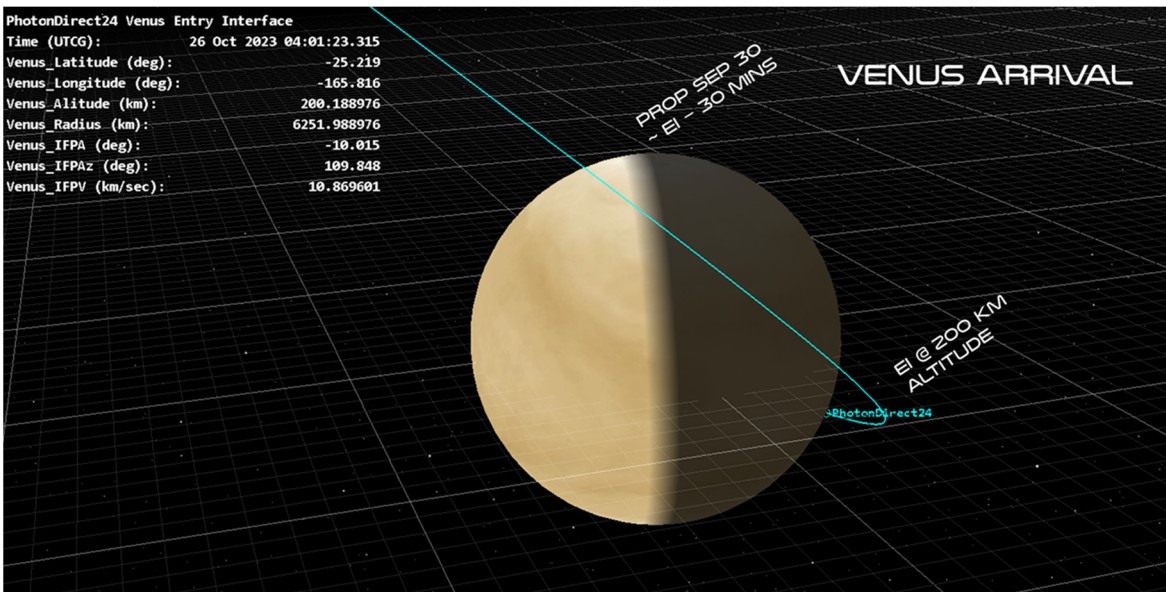

**Figure 5.** High-energy Photon bus releases the entry probe 30 min before entry interface (EI) after targeting the entry interface selected for optimal instrument measurement conditions.

## 4. The Probe

The small probe (Figure 6) will contain up to 1 kg of science payload to search for organic chemicals in the cloud particles and explore the habitability of the clouds, achieving ~330 s in the cloud layer between ~45–60 km altitude to perform science operations. The science instrument is an autofluorescing nephelometer (AFN) described in [3]. The small probe is a ~40 cm diameter, 45-degree half-angle sphere-cone blunt body with a hemispherical aft body for static stability in the hypersonic flow regime [4].

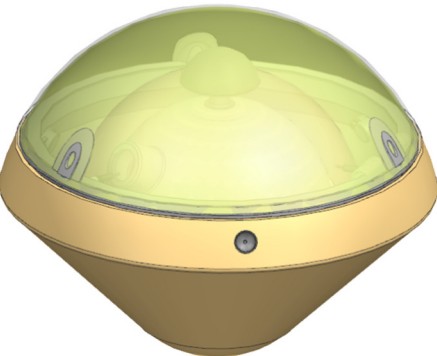

**Figure 6.** The small Venus probe is a 45-degree half-angle sphere cone ~40 cm in diameter.

The probe shape was traded based on the stability characteristics in various flow regimes (hypersonic, transonic, subsonic, etc.) and center of gravity location constraints, among other considerations.

The probe diameter was selected to accommodate a pressure vessel as well as the instrument payload, considering the required focal length of the nephelometer and the size of the on-board systems. Housing the probe electronics in a pressure vessel allows for a robust overall design. The aluminum pressure vessel contains all of the system components with the exception of thermometers, pressure sensors, and the probe antenna and is surrounded by a structural insulation layer. The insulation maintains the flight computer, radio, and instrument at a suitable operating pressure, acting as a thermal sink to maintain allowable operating temperatures, and act as a barrier to the corrosive Venusian atmosphere.

The pressure vessel wall thickness is driven by three primary considerations: the mass of material needed to absorb the thermal load from both the internal components and the Venusian environment, the pressure the vessel must withstand to enable transmission of science data for the required amount of time once through the cloud layer as pressure and temperature build, and the manufacturing methods. With a 2 mm baseline thickness, the driving constraints are manufacturing best practices, providing some margin against increases in the thermal, power, and data budgets.

The probe forebody TPS material is either Heat-shield for Extreme Entry Environment (HEEET) or carbon phenolic with the aftbody TPS material a radio frequency (RF) transparent and acid-resistant Polytetrafluoroethylene (PTFE, e.g., Teflon™).

## 5. Concepts of Operations

The probe will follow the following sequence of events during the Science Phase (Figure 7), with absolute timing dependent upon the selected EFPA (−10 degree baseline shown):

- Probe release and spin up after final entry interface targeting;
- Coast phase (~2 h, low energy state);
- Pre-entry (initialization of key systems, TBD timing);
- Relay communications begins and continues throughout science phase;
- Entry interface reached;
- Heating pulse, RF blackout, peak G's (40–80 s after entry interface);
- Enter clouds (180 s after entry interface);
- Primary science data collection (330 s data collection);
- Leave clouds (520 s after entry interface);
- Continued data transmission/re-transmission of science data (~20 min duration);
- Pressure vessel design limit reached, expected LOS (~30 min after entry interface);
- Surface contact (~3500–4000 s after entry interface).

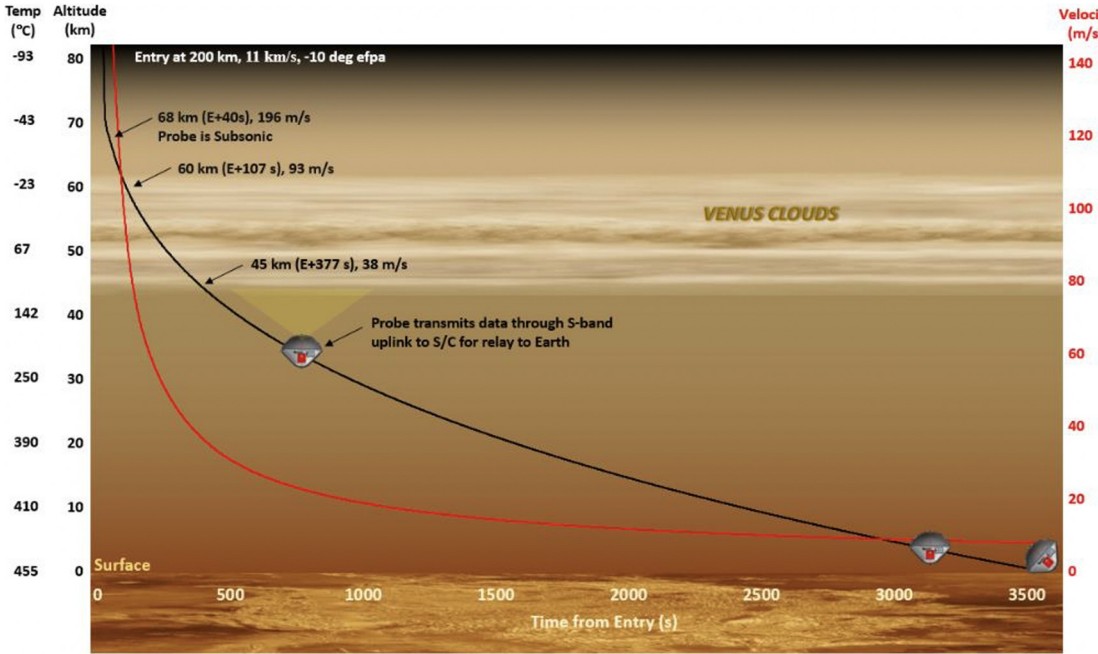

**Figure 7.** The science phase targets the Venus cloud layer between 45 and 60 km altitude, enabling ~330 s of science observations (Credit: NASA ARC).

Through the cloud layer and below, the science data will be transmitted direct to Earth at optimized data rates. Objectives below the cloud layer, such as the potential to continue science observations with the primary instrument or to return environmental data will be performed on a best effort basis only.

## 6. Summary of Science Goals for the Rocket Lab Mission

The mission is the first opportunity to probe the Venus cloud particles directly in nearly four decades. Even with the mass and data rate constraints and the limited time in the Venus atmosphere, breakthrough science is possible. We have chosen a low-mass, low-cost autofluorescing nephelometer (AFN) to meet the Rocket lab Mission science objectives [3].

The overarching science goals are the search for evidence of life or habitability in the Venusian clouds. There are two specific science objectives: to search for the presence of organic molecules within cloud-layer particles and to determine the shape and indices of refraction (a proxy for composition) of the Mode 3 cloud particles. See [3] for the detailed description of the AFN instrument development. For discussion of the motivation and overall science objectives of the Venus Life Finder (VLF) missions, see [5].

**Author Contributions:** Writing—original draft preparation, R.F. and C.M.; writing—review and editing, R.F., C.M., R.H., E.M., D.S., P.B., S.S., J.J.P., C.E.C., D.H.G. and D.B. All authors have read and agreed to the published version of the manuscript.

**Funding:** Rocket Lab.

**Institutional Review Board Statement:** Not applicable.

**Informed Consent Statement:** Not applicable.

**Data Availability Statement:** Not applicable.

**Acknowledgments:** We thank the Rocket Lab Venus Team and the Venus Life Finder (VLF) Mission team for useful discussions. Individuals involved as the Rocket Lab Venus Team can be found here: https://www.rocketlabusa.com/, accessed on 12 August 2022.

**Conflicts of Interest:** The authors declare no conflict of interest.

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
