# Peer review of "Rocket Lab Mission to Venus"

_aerospace, doi:10.3390/aerospace9080445_

Round 1

Reviewer 1 Report

"Rocket Lab Mission to Venus" is a fascinating manuscript and, I believe, a soon-to-be published article. The paper presents a groundbreaking private science mission to Venus. If the mission will succeed, the history of space exploration will be rewritten.

It is obvious that authors have carefully considered what and how to present in the paper. The rocket and spacecraft have a recent flight experience to the moon which requires many 'deep space' functions. I think that shows the feasibility of the Rocket Lab team to carry out the proposed, analysed and developed Venus mission.

The paper describes the high-energy Photon spacecraft, its integration with the Electron rocket and the thrust capabilities. The trajectory is presented from multiple aspects: timeline, tradeoffs and graphics. I think there is enough detail to reproduce results in commonly available mission design tools. The scientific probe section describes the most important design decisions which serves as an overview of the instrument and its protective enclosure. The concept of operations is brief providing the overall idea which I think is acceptable for this paper. If anything, ConOp section can be expanded but I'm afraid it would require several pages to provide the next level of detail (editors can decide if its necessary or not, depending on how it fits with the rest of the special issue). Instead of conclusions, the paper provides a short summary. This is acceptable because the real conclusion is "there is more work to do". Good luck and thank for the exciting paper!

All details are to the point and concise with further references if the reader is interested. I think this is a perfect landmark paper giving an update of the project while the team is preparing for the May 2023 launch. 

Author Response

We provide responses to the Reviewers’ comments on the Aerospace-1852680 manuscript. Our response is marked in bold font and the reviewers’ comments are in regular font.

Response to the Reviewer 1:

"Rocket Lab Mission to Venus" is a fascinating manuscript and, I believe, a soon-to-be published article. The paper presents a groundbreaking private science mission to Venus. If the mission will succeed, the history of space exploration will be rewritten.

It is obvious that authors have carefully considered what and how to present in the paper. The rocket and spacecraft have a recent flight experience to the moon which requires many 'deep space' functions. I think that shows the feasibility of the Rocket Lab team to carry out the proposed, analysed and developed Venus mission.

The paper describes the high-energy Photon spacecraft, its integration with the Electron rocket and the thrust capabilities. The trajectory is presented from multiple aspects: timeline, tradeoffs and graphics. I think there is enough detail to reproduce results in commonly available mission design tools. The scientific probe section describes the most important design decisions which serves as an overview of the instrument and its protective enclosure. The concept of operations is brief providing the overall idea which I think is acceptable for this paper. If anything, ConOp section can be expanded but I'm afraid it would require several pages to provide the next level of detail (editors can decide if its necessary or not, depending on how it fits with the rest of the special issue). Instead of conclusions, the paper provides a short summary. This is acceptable because the real conclusion is "there is more work to do". Good luck and thank for the exciting paper!

All details are to the point and concise with further references if the reader is interested. I think this is a perfect landmark paper giving an update of the project while the team is preparing for the May 2023 launch. 

Thank you for reviewing and accepting our manuscript.

Reviewer 2 Report

This is an overview of the first and simplest of three Venus Life Finder (VLF) missions designed to assess the habitability of the Venutian atmosphere.  It describes a low-cost mission to directly insert a one-way probe to detect organic compounds, if present, in the cloud layers of Venus and to characterize the cloud particles thought most likely to contain living organisms, if they exist in that environment.

The VLF mission concepts are the product of a multi-year study by an international consortium of experts coordinated by Sara Seager and Janusz J. Petkowski at MIT.  The core team includes David Grinspoon, among the earliest and most persistent advocates of exploring the potential for life in the lower cloud layers of Venus, along with other experts well qualified to plan such a mission, in concert with the Rocket Lab which is committed to flying the mission in 2023. The rationale for the VLF missions and much more technical detail is given in the report (December 2021) of that group, which now is publishing different elements of the VLF missions in piecemeal fashion.  This paper provides a nice overview of the first VLF mission, while technical details have or will appear in other publications.

Lower-cost, low payload missions of this type seem likely to be the major way in which exploration of planetary bodies in our solar system is likely to proceed through at least the medium-term future.  As such, publication of this paper is timely and important.  All the authors are well-qualified in their respective areas.  The mission design and its capabilities are well described.  I draw the authors' attention to three minor points.

Line 109 -- Presumably "70,000" is intended instead of "70,0000".

Fig. 4 -- Some of the lettering in the figure is too small for clarity.

Fig. 5 -- The meaning of this figure is not clear. Neither "pericythe" nor "scout" is mentioned in the text.  The two blue lines extending from Earth are not explained.  Perhaps to astrophysicists the figure is self-evident, but without a more elaborate legend, its usefulness is questionable.  

Line 209 -- Either the sentence should be reworded, or a colon should replace "are."

Author Response

We provide responses to the Reviewers’ comments on the Aerospace-1852680 manuscript. Our response is marked in bold font and the reviewers’ comments are in regular font.

Response to the Reviewer 2:

This is an overview of the first and simplest of three Venus Life Finder (VLF) missions designed to assess the habitability of the Venutian atmosphere.  It describes a low-cost mission to directly insert a one-way probe to detect organic compounds, if present, in the cloud layers of Venus and to characterize the cloud particles thought most likely to contain living organisms, if they exist in that environment.

The VLF mission concepts are the product of a multi-year study by an international consortium of experts coordinated by Sara Seager and Janusz J. Petkowski at MIT.  The core team includes David Grinspoon, among the earliest and most persistent advocates of exploring the potential for life in the lower cloud layers of Venus, along with other experts well qualified to plan such a mission, in concert with the Rocket Lab which is committed to flying the mission in 2023. The rationale for the VLF missions and much more technical detail is given in the report (December 2021) of that group, which now is publishing different elements of the VLF missions in piecemeal fashion.  This paper provides a nice overview of the first VLF mission, while technical details have or will appear in other publications.

Lower-cost, low payload missions of this type seem likely to be the major way in which exploration of planetary bodies in our solar system is likely to proceed through at least the medium-term future.  As such, publication of this paper is timely and important.  All the authors are well-qualified in their respective areas.  The mission design and its capabilities are well described.  I draw the authors' attention to three minor points.

We thank the reviewer for accepting our manuscript.

Line 109 -- Presumably "70,000" is intended instead of "70,0000".

Corrected.

Fig. 4 -- Some of the lettering in the figure is too small for clarity.

Noted, however we have decided to keep the figure at original dimensions, which is a standard Rocket Lab figure design and font.

Fig. 5 -- The meaning of this figure is not clear. Neither "pericythe" nor "scout" is mentioned in the text.  The two blue lines extending from Earth are not explained.  Perhaps to astrophysicists the figure is self-evident, but without a more elaborate legend, its usefulness is questionable.  

We have replaced Figure 5 with a new simpler and more up to date figure that should also be more clear.

Line 209 -- Either the sentence should be reworded, or a colon should replace "are."

Corrected.